# CYSTATIN C—A Monitoring Perspective of Chronic Kidney Disease in Patients with Diabetes

**DOI:** 10.3390/ijms25158135

**Published:** 2024-07-26

**Authors:** Alexandra-Mihaela Visinescu, Emilia Rusu, Andrada Cosoreanu, Gabriela Radulian

**Affiliations:** 1Department of Diabetes, Nutrition and Metabolic Diseases, “Carol Davila” University of Medicine and Pharmacy, 37 Dionisie Lupu Street, 030167 Bucharest, Romania; alexandra-mihaela.visinescu@drd.umfcd.ro (A.-M.V.); andrada.cosoreanu@drd.umfcd.ro (A.C.); gabriela.radulian@umfcd.ro (G.R.); 2Department of Diabetes, Nutrition and Metabolic Diseases, “Prof. Dr. N. C. Paulescu” National Institute of Diabetes, Nutrition and Metabolic Diseases, 5-7 Ion Movila Street, 020475 Bucharest, Romania; 3Department of Diabetes, “N. Malaxa” Clinical Hospital, 12 Vergului Street, 022441 Bucharest, Romania

**Keywords:** diabetes mellitus, serum creatinine, serum cystatin C, estimated glomerular filtration rate, chronic kidney disease

## Abstract

Chronic kidney disease (CKD) is a microvascular complication that frequently affects numerous patients diagnosed with diabetes. For the diagnosis of CKD, the guidelines recommend the identification of the urinary albumin/creatinine ratio and the determination of serum creatinine, based on which the estimated rate of glomerular filtration (eGFR) is calculated. Serum creatinine is routinely measured in clinical practice and reported as creatinine-based estimated glomerular filtration rate (eGFRcr). It has enormous importance in numerous clinical decisions, including the detection and management of CKD, the interpretation of symptoms potentially related to this pathology and the determination of drug dosage. The equations based on cystatin C involve smaller differences between race groups compared to GFR estimates based solely on creatinine. The cystatin C-based estimated glomerular filtration rate (eGFRcys) or its combination with creatinine (eGFRcr-cys) are suggested as confirmatory tests in cases where creatinine is known to be less precise or where a more valid GFR estimate is necessary for medical decisions. Serum creatinine is influenced by numerous factors: age, gender, race, muscle mass, high-protein diet, including protein supplements, and the use of medications that decrease tubular creatinine excretion (H2 blockers, trimethoprim, fenofibrate, ritonavir, and other HIV drugs). The low levels of creatinine stemming from a vegetarian diet, limb amputation, and conditions associated with sarcopenia such as cirrhosis, malnutrition, and malignancies may lead to inaccurately lower eGFRcr values. Therefore, determining the GFR based on serum creatinine is not very precise. This review aims to identify a new perspective in monitoring renal function, considering the disadvantages of determining the GFR based exclusively on serum creatinine.

## 1. Introduction

CKD is a complex condition prevalent in both type 1 and type 2 diabetes [1]. The International Diabetes Federation specifies that 537 million adults aged 20–79 are presently living with diabetes, a number estimated to rise to 784 million cases by 2045. Additionally, owing partly to the escalation in risk factors like obesity and diabetes, the incidence of CKD has increased globally, impacting an estimated 850 million individuals [2]. According to data published in 2021, CKD is 10 times more common in patients diagnosed with diabetes [3]. Approximately 10% of adults globally [4,5] and 14% of adults in the United States are affected by CKD [4]. Statistics from the US National Health and Nutrition Examination Survey up to 2012 revealed evidence of CKD based on eGFR and/or urinary albumin/creatinine ratio in roughly 40% of individuals with type 2 diabetes and approximately 60% of people aged 65 or older with type 2 diabetes [4,6]. Additionally, the reported prevalence of CKD among individuals with type 2 diabetes globally stands at around 50% [4,7].

CKD is characterized by multifactorial pathogenesis involving tubular, glomerular, and inflammatory changes, ultimately culminating in irreversible renal fibrosis. Chronic hyperglycemia triggers numerous pathological processes that involve glomerular endothelial cells, mesangial cells, podocytes, and smooth muscle cells. CKD manifests through multiple developmental stages with hyperglycemia-induced metabolic and hemodynamic pathways widely acknowledged as key mechanisms driving its progression [8]. 

The diagnostic indicators of “nephropathy”, such as the presence of albuminuria and/or a diminished eGFR are not optimal from a clinical standpoint, as many of them become abnormal only after a considerable decline in renal function [9,10,11]. While kidney biopsy remains the gold standard for distinguishing between diabetic and non-diabetic nephropathy [9,12], its invasive character makes it unsuitable for routine clinical practice. Diabetic nephropathy and nephroangiosclerosis stemming from arterial hypertension stand as the leading causes of CKD. The escalating prevalence of common risk factors like diabetes, hypertension, obesity, and glomerular diseases in high-income nations coupled with infections prevalent among impoverished populations will further exacerbate the already substantial burdens associated with CKD. The extent of segmental and global glomerulosclerosis, interstitial fibrosis, and tubular atrophy determines chronicity in kidney disease. At an advanced stage, the disease is typified by a sustained state of hypoxia, oxidative stress, and inflammation, all of which contribute to the development of renal fibrosis, a distinct and irreversible hallmark of all CKD [1]. 

## 2. Serum Creatinine and/or Cystatin C? 

### 2.1. CKD Is Underdiagnosed

CKD is significantly underdiagnosed, especially in the early stages in patients with diabetes. The ADD-CKD Study (Awareness, Detection and Drug Therapy in Type 2 Diabetes and Chronic Kidney Disease) is a retrospective, observational study conducted over 15 months (9037 adult patients with type 2 diabetes). Out of 5036 patients with CKD, only 607 (12.1%) were diagnosed. This study evaluated the percentage of undiagnosed CKD in patients with diabetes (Table 1) [13]. 

### 2.2. Markers in the Monitoring of Renal Function

The best indicator for the evaluation of kidney function is the GFR. It is defined as the rate of flow in milliliters per minute of plasma from which substances are freely filtered by the membranes of the kidney glomeruli [14,15]. The gold standard method for its assessment is renal inulin clearance. Using exogenous substances like inulin or radioactive markers for measuring GFR is invasive and expensive, potentially resulting in serious complications and high costs. Nevertheless, inulin is not suitable for daily practice [14,16]. Endogenous markers utilized in estimating GFR include cystatin C and serum creatinine [14,15]. 

The diagnosis of CKD is based on the presence of either of two criteria persisting for three months or more: the measurement of renal function assessed using the updated race-free Chronic Kidney Disease Epidemiology (CKD-EPI) 2021 formula, indicating an eGFR below 60 mL/min/1.73 m^2^ and/or albuminuria (urinary albumin/creatinine ratio) exceeding 30 mg/g [4]. 

CKD is classified according to GFR values, as in the following table (Table 2) [17]: 

At the last stage, known as end-stage renal disease, kidney function is so severely diminished that most patients require renal replacement therapy (RRT) to survive. RRT includes peritoneal dialysis, hemodialysis, or kidney transplantation. The transplant option is preferred due to its better quality of life, improved patient survival rates, and lower costs compared to dialysis treatments [18]. 

Table 3 indicates the classification of CKD depending on the albuminuria categories [17].

Creatinine originates as a breakdown product of creatine phosphate in skeletal muscle. It is derived from both a patient’s inherent muscle metabolism and the intake of dietary creatine found abundantly in sources like meat and creatine supplements. Although it undergoes free filtration by the renal glomeruli without reabsorption or renal metabolism, it is actively secreted by the proximal tubule. This active secretion significantly contributes to the urinary creatinine content, potentially leading to an overestimation of the GFR calculated through 24-h creatinine clearance compared to the GFR measured by the gold standard method, inulin clearance, by a margin of 10 to 40% [19,20]. 

Cystatin C is a nonglycosylated basic protein produced by all nucleated cells [9,21]. Cystatin C, a cysteine protease inhibitor from the type 2 cystatin gene family, is encoded by the CST3 gene. The primary function of cystatins is to regulate endogenous proteinase activity, typically secreted or leaked from lysosomes of dying or diseased cells. It was first identified in cerebrospinal fluid in 1961 and its complete amino acid sequence was established in 1981. This non-glycosylated protein consists of 120 amino acids and has a molecular mass of 13,343 Da. At position 11 of the N-terminus, cystatin C features a conserved glycine common to all cystatins. The N-terminal segment is crucial for both binding affinity and inhibition specificity. Truncating the N-terminus of human cystatin C by 10 peptides significantly reduces its inhibition of cathepsin H and cathepsin B by over 50-fold and 1000-fold, respectively [18]. 

Cystatin C undergoes filtration by the renal glomeruli, followed by complete reabsorption and metabolism in the proximal tubule, with no excretion in the urine [22]. It has been integrated into routine clinical practice in Sweden since 1995 [23]. In 2012, the Kidney Disease Improving Global Outcomes (KDIGO) guideline recommended the measurement of cystatin C in adults with eGFRcr between 45–60 mL/min/1.73 m^2^, particularly in cases where other indicators of kidney damage are absent to validate the diagnosis of CKD [19]. Cystatin C has been recognized for over 60 years and the Food and Drug Administration approved a cystatin C assay for estimating the GFR in 2001. Unlike creatinine, cystatin C levels are not influenced by race or muscle mass. However, it can be elevated by factors such as inflammation, smoking, obesity, thyroid disease, steroid therapy, and chemotherapy. These characteristics make cystatin C an alternative and supplement to creatinine in estimating the GFR [4]. While the cost of measuring cystatin C may be higher than measuring creatinine [4,24], the importance for individuals with diabetes to accurately determine renal function and to classify CKD and the cardiovascular risk justifies the utilization of cystatin C with creatinine for estimating eGFR [4].

Fever, infection, menstruation, exercise within 24 h, congestive heart failure, marked hyperglycemia, or hypertension may increase urinary albumin-creatinine ratio separate from kidney damage [25]. Multiple studies have determined that not all patients with diabetic kidney disease exhibit elevated urinary albumin-creatinine ratio values in the early stages of the condition, suggesting that it may not be sufficiently sensitive as a marker during the initial phases [26,27]. Serum creatinine concentration rises only when roughly 40–50% of the kidney parenchyma is impaired. Conversely, albuminuria often precedes a decline in GFR, but it may be absent in cases of tubulointerstitial or hypertensive kidney diseases [28]. Additionally, it is worth noting that approximately 30% of patients with diabetic kidney disease exhibit normal urine albumin levels [1,29].

### 2.3. Equations for Estimating GFR

The European Federation of Clinical Chemistry and Laboratory Medicine maintains its support for the CKD-EPI 2009 equation without reporting values specifically for Black individuals due to the lower representation of Black people in European countries. However, it also recommends an increased utilization of cystatin C due to its independence from race, aiming for more accurate GFR estimates across diverse populations [23,30]. 

The 2009 CKD-EPI creatinine equation includes sex, age, creatinine, and race [17,31]. On the other hand, the 2021 CKD-EPI creatinine equation includes age, sex, and creatinine, excluding race [17,32]. As a result of not including the coefficient for the Black race, the 2021 CKD-EPI creatinine equation tends to slightly overestimate GFR in the non-Black people and underestimate it in the Black population. In the non-Black race group, the 2009 CKD-EPI creatinine equation is more accurate than the 2021 CKD-EPI creatinine equation [17]. The initial publication of the new CKD-EPI equations, both for cystatin C alone and in combination with creatinine, occurred in 2012 [19,33]. 

The American Diabetes Association specified in the 2012 guideline that eGFRcr, estimated using the Modification of Diet in Renal Disease (MDRD) or Chronic Kidney Disease Epidemiology Collaboration (CKD-EPI) formulas, can be employed for evaluating GFR in patients with CKD [14,34]. The most recent guideline of the American Diabetes Association published in 2024 mentioned that the Refit 2021 CKD-EPI equation is now recommended for everyone for estimation of GFR [25]. 

According to the KDIGO 2024 guideline, the use of eGFRcr is proposed for adults at risk of CKD. However, if cystatin C is accessible, the GFR category should be estimated using the eGFRcr-cys. The Work Group determined that many individuals at risk for CKD would prioritize accuracy when confirming the diagnosis and staging of CKD. This recommendation places a low value on the availability and cost of assessing eGFRcr-cys, indicating that individuals at risk of CKD would prefer the more accurate assessment despite any associated costs or availability concerns. In the CKD-PC collaboration, where 720,736 individuals had measures of cystatin C in addition to eGFRcr and urinary albumin/creatinine ratio, replacing the assessment of eGFRcr with eGFRcr-cys in the matrix of GFR categories resulted in several changes in risk distributions. Notably, the group with an eGFR category of 45–59 mL/min/1.73 m^2^ and urinary albumin/creatinine ratio <10 mg/g (<1 mg/mmol) was shifted to higher risk for all ten outcomes. As a result, this category was no longer classified as low risk (“green”) for any of the complications. The data from KDIGO suggests that the combined eGFRcr-cys equation is more beneficial for differentiating GFR risk stages in comparison with eGFRcr alone. While GFR is a component of excretory function, it is widely acknowledged as the premier index of kidney function. The kidneys undertake several essential roles in the body, encompassing the metabolism and excretion of substances, production of erythropoietin, regulation of volume and blood pressure, and maintenance of electrolytes, acid-base balance, and mineral homeostasis. Glomerular filtration is indeed one of the vital functions performed by the kidney, but it is just one aspect of its multifaceted role. The greater accuracy of eGFRcr-cys compared with eGFRcys or eGFRcr is consistently determined in studies evaluating GFR estimating equations in comparison with mGFR across various countries such as Congo, Brazil, China, Singapore, Japan, and Pakistan. In these studies, the P30 is estimated to be between 80%-90%, which is considered optimal for many purposes [17]. In two large-scale studies conducted across pooled cohorts of both general population and clinical populations in Europe and North America, the P30 (described as the percentage of the values of eGFR within 30% of mGFR) using eGFRcr-cys falls within the range of 90%, which is considered optimal [17,35].

Two US National Kidney Disease Organizations propose the more frequent use of both the GFR estimating equations with cystatin C and race-free equations, because those based on cystatin C involve smaller differences between race groups compared to GFR estimates based solely on creatinine [23,36]. The eGFR using creatinine determines day-to-day variability between 4%–10% in healthy people and between 5%–15% in individuals with CKD [4,24].

### 2.4. eGFRdiff—A Beneficial Indicator 

The routine screening for cystatin C has been strongly recommended by the American Society of Nephrology and the US National Kidney Foundation. He et al. conducted a prospective cohort study involving 25,825 participants with diabetes who were free of diabetic microvascular complications (DMCs), diabetic retinopathy, diabetic neuropathy, and diabetic kidney disease at baseline (2006–2010) from the UK Biobank (22 sites across Scotland, England, and Wales). The eGFR difference (eGFRdiff) between eGFRcr and eGFRcys was calculated using both the absolute difference (eGFRabdiff) defined as eGFRcys minus eGFRcr and the relative difference (eGFRrediff) defined as eGFRcys/eGFRcr between creatinine- and cystatin C-based estimations. A large eGFRdiff was independently associated with an increased risk of DMCs. Over a median follow-up period of 13.6 years, DMCs manifested in 5753 participants, comprising 2752 cases of diabetic retinopathy, 3203 cases of diabetic kidney disease, and 1149 cases of diabetic neuropathy. The significant variation within individuals between eGFRcr and eGFRcys has become increasingly apparent in recent years and it is a cause for concern. This variation has been reported to be linked with various adverse events, such as hospital admissions, falls, cardiovascular events, kidney failure, and mortality. Monitoring eGFRdiff among populations diagnosed with diabetes could offer significant prognostic insights for risk stratification and early intervention. The study suggested that eGFRdiff serves as an early indicator for DMCs and indicated that monitoring eGFRdiff in the population with diabetes could be beneficial for identifying high-risk patients [37].

### 2.5. Indications for the Use of Cystatin C

The current KDIGO guideline proposes the following indications for the use of cystatin C: -*Diet*: low-protein diet, keto diets, vegetarian, high-protein diets, and creatine supplements;-*llness other than CKD*: malnutrition, heart failure, catabolic consuming diseases, cancer, cirrhosis, muscle wasting diseases;-*Lifestyle*: smoking;-*Medication effects*: broad spectrum antibiotics that minimize extrarenal elimination, decreases in tubular secretion, steroids (anabolic, hormone);-*Variations in body habitus and muscle mass*: eating disorders, extreme sports or exercise, bodybuilding, above-knee amputation, spinal cord injuries resulting in paraplegia, paraparesis, quadriplegia or quadriparesis, and class III obesity [17].

### 2.6. Limitations in Using Cystatin C 

The 2019 General Chemistry Survey by the College of American Pathologists indicated that only 7% of US clinical laboratories offered cystatin C tests and over 90% of these referred the testing to external commercial labs [24]. 

Cystatin C testing costs around GBP 2.50 (USD 3.00) per test, which is about 10 times higher than the GBP 0.25 (USD 0.30) cost of creatinine testing [38]. 

Cystatin C measurement costs more than creatinine testing, but it is less expensive than other common laboratory tests, such as brain natriuretic peptide, troponin, parathyroid hormone, or vitamin D and significantly cheaper than mGFR testing [23]. 

There are some limitations in using cystatin C, as indicated in Table 4 [24]. 

## 3. Novel Biomarkers in Diabetic Kidney Disease

Diabetic kidney disease is a heterogeneous condition with a complex pathophysiology. Therefore, it is unlikely that a single biomarker can predict its prognosis. A multi-marker approach may be necessary to predict disease progression [39].

The classification of novel biomarkers is presented in Table 5 [8]. 

## 4. Discussions

### 4.1. Cystatin C—An Earlier Marker in the Diagnosis of CKD than Albuminuria

In a meta-analysis conducted by Arceo et al. in 2019, it was demonstrated that cystatin C serves as a cost-effective biomarker with superior utility for the early diagnosis of CKD [40,41]. Individuals with type 2 diabetes often transition through a prediabetic phase and may already exhibit renal dysfunction at the time of diagnosis. While microalbuminuria has traditionally been regarded as the earliest clinical indicator of CKD, the studies indicate that a significant proportion of individuals with type 2 diabetes (ranging from 29.1% to 61.6%) may experience renal impairment even before the onset of microalbuminuria, which is considered the gold standard for early diagnosis [42].

Jeon et al. examined MDRD, CKD-EPI-creatinine, and cystatin C levels in patients with type 2 diabetes in association with a normal level of albuminuria (n = 332), microalbuminuria (n = 83), and macroalbuminuria (n = 42). Similar to the study conducted by Kadriye Akpınar et al., MDRD and CKD-EPI eGFR were found to be significantly lower in the groups with macroalbuminuria and microalbuminuria compared to the group with normoalbuminuria. The levels of cystatin C in both serum and urine showed an increase corresponding to the degree of albuminuria. In the study by Kedam et al., a total of 239 patients with type 2 diabetes were evaluated, including 110 with normoalbuminuria, 81 with microalbuminuria, and 48 with macroalbuminuria. The levels of serum cystatin C were found to be negatively associated with MDRD eGFR and were notably higher in the group with macroalbuminuria compared to the groups with normoalbuminuria and microalbuminuria. However, there was no significant difference in serum cystatin C levels between the groups with normoalbuminuria and microalbuminuria. These results may be attributed to the relatively short and similar durations of diabetes mellitus in the groups with normoalbuminuria and microalbuminuria (ranging from 5.0 to 7.5 years). A longer duration of diabetes mellitus is known to be a factor that increases cystatin C levels, potentially leading to renal damage. The authors mentioned that the levels of urinary and serum cystatin C serve as useful markers for renal dysfunction in type 2 diabetes associated with normoalbuminuria [14].

### 4.2. Variations of Serum Creatinine and Cystatin C in Different Clinical Conditions

Cardiovascular disease is the primary cause of complications and mortality in patients with type 2 diabetes mellitus. This is linked to prolonged hyperglycemia, causing vascular damage and accelerating the development and progression of arteriosclerosis. Patients diagnosed with CKD have a high risk of cardiovascular disease. Besides traditional risk factors like hypertension, smoking, and aging, specific CKD-related factors such as uremic toxins (including advanced glycation end-products phosphate, uric acid, and endothelin-1) and mineral metabolism disorders can cause arterial stiffness and decreased vascular compliance through chronic inflammation and oxidative stress. A cumulative mortality survey involving 15,046 participants found that the 10-year cumulative standardized cardiovascular disease mortality rate for individuals with both type 2 diabetes mellitus and CKD was 19.6%. This rate was considerably higher compared to the general population (3.4%), patients with only diabetes (6.7%), and patients with only CKD (9.9%) [43]. 

Briet et al. found that arterial stiffness increases as renal function declines in patients with mild-to-moderate CKD [44]. 

Cystatin C serves as a sensitive marker of kidney function, providing a rapid and accurate reflection of the endogenous glomerular filtration rate. Luo et al. indicated that cystatin C has a positive correlation with brachial-ankle pulse wave velocity (r = 0.448, *p* < 0.001) [43]. 

The association between brachial-ankle pulse wave velocity and cystatin C was stronger in young subjects compared to middle-aged and older patients. An explanation could be that the risk of cardiovascular disease is lower in young individuals and brachial-ankle pulse wave velocity has a stronger association in those with a lower cardiovascular disease risk. Luo et al. demonstrated that cystatin C represents an independent indicator of peripheral arterial stiffness (defined as a brachial-ankle pulse wave velocity ≥ 1800 cm/s) in patients with type 2 diabetes combined with CKD [43]. 

Yeli Wang et al. offer guidance on interpreting discordant eGFRcr and eGFRcys results in ambulatory adults [23]. 

Approximately 30% of participants had a difference greater than 15 mL/min/1.73 m^2^ and in most cases eGFRcr was higher than eGFRcys. In the situations where eGFRcr and eGFRcys were discordant, eGFRcr-cys provided the most accurate estimate of mGFR. A significant difference between eGFRcr and eGFRcys indicates a substantial error compared to mGFR in either eGFRcr, eGFRcys, or both. Typically, these errors are based on clinical conditions that impact the creatinine or cystatin C levels independently of GFR. Non-GFR determinants include factors such as tubular secretion or reabsorption and extra-renal elimination of cystatin C or creatinine [23]. 

Other studies have shown that inactivity, muscle wasting, and malnutrition are linked to lower serum creatinine levels resulting in a higher eGFRcr relative to mGFR. Conversely, factors such as chronic inflammation (as indicated by insulin resistance), obesity, smoking, elevated levels of tumor necrosis factor and C-reactive protein, or lower serum albumin levels have been linked with higher serum cystatin C levels, leading to lower eGFRcys according to mGFR [23,45]. 

Another hypothesis regarding the cause of significant differences between eGFRcys and eGFRcr is that eGFRcys might be falsely low due to selective damage to large pores in the wall of the glomerular capillary in charge of the filtration of cystatin C, a phenomenon sometimes referred to as “shrunken pore syndrome”. This condition increases the cardiovascular risk. Certainly, in cases where significant negative differences exist between eGFRcys and eGFRcr, the utilization of eGFRcr-cys resulted in notable reductions in large errors in GFR estimation, decreasing from 30% (P30 of 70%) to 12% (P30 of 88%), representing an important 60% reduction in larger errors [23]. Grubb et al. proposed this hypothesis about the "shrunken pore syndrome" that describes a pathophysiologic condition characterized by an eGFRcys/eGFRcr ratio of less than 0.6 or 0.7. It reflects selective impairment in the glomerular filtration of cystatin C and other molecules of similar molecular weight (5–30 kDa). Two proteomic studies have shown that certain serum proteins, such as MCP-3, interleukin 6 and osteoprotegerin, accumulate in patients with this syndrome. These proteins are believed to be linked to inflammation, atherosclerosis, and vascular endothelial injury, all of which are significant causative factors for DMCs [37]. 

Dejenie et al. found significant abnormalities in serum cystatin C and lipoprotein levels in patients with type 2 diabetes and diabetic nephropathy compared to those with normal renal function. A lot of experimental studies have explained the mechanism by which hyperlipidemia can cause glomerulosclerosis and tubulointerstitial sclerosis. It stimulates resident renal cells to produce fibrotic cytokines and chemokines, causing the infiltration of macrophages and monocytes into glomerular tissue and promoting extracellular matrix deposition [9]. 

More significant accuracy for eGFRcr-cys compared to either eGFRcys or eGFRcr was identified in individuals with obesity, cancer, or HIV. Supporting these findings, a large study of individuals living in Sweden, specifically Stockholm, who were referred for a mGFR test and had diagnoses of diabetes, heart and liver failure, cancer, or cardiovascular disease, found eGFRcr-cys to be the most precise and least biased estimator of GFR [17]. 

The impact of consuming cooked meat or fish meal on creatinine concentrations has been evaluated [46]. For instance, a study illustrated an increase in serum creatinine of around 20 μmol/L (approximately 0.23 mg/dL), which in the study population was similar to a decline in eGFR of around 20 mL/min/1.73 m^2^. Maximum postprandial effects were observed in some individuals after 2 h while others experienced them after 4 h. To mitigate this effect, it is recommended, after the intake of fish or meat, to wait for at least 12 h before measuring serum creatinine. However, the KDIGO guideline appreciates that implementing this approach in clinical settings may be challenging [17]. 

In a study that compared eGFRcys and eGFRcr before and after amputation in military veterans, a significant change was observed in eGFRcr, as expected with the lack of a limb and mobility. However, there was no change in eGFRcys [17,47]. 

Serum cystatin C levels exhibited a stronger correlation with measured mGFR compared to serum creatinine levels in a study involving individuals with anorexia [17,48]. 

Another study showed that factors associated with increased values for eGFRcr in comparison with eGFRcys included female sex, non-Black race, older age, current smoking, higher BMI, weight loss, and higher eGFR [17,49]. 

Furthermore, in context involving substantial weight loss, such as those observed post-bariatric surgery and with incretin-based diabetes treatment strategies, around 25% of the lost weight comprises skeletal muscle loss. Although not directly explored, it is speculated that the utilization of eGFR based solely on creatinine in this case could potentially overestimate renal function [4]. 

In a post hoc analysis that included data pooled from nine phase 3, randomized clinical trials involving 4745 patients with type 2 diabetes being treated with dapagliflozin and having an eGFRcr between 30–60 mL/min/1.73 m^2^ (Stage 3 CKD), the inclusion of eGFRcys led to a reclassification of 66% of patients as having an eGFR > 60 mL/min/1.73 m^2^ [50]. 

Moreover, in the meta-analysis of 11 population studies performed by Shlipak M.G. et al. comprising 90,750 participants, contrasting results were reported: 42% of patients with CKD stage 3a (eGFR creatinine 45–59 mL/min/1.73 m^2^) were reclassified as having an eGFR > 60 mL/min/1.73 m^2^ when utilizing eGFRcys [51]. 

Researchers examined the impact of cystatin C measurement on CKD classification in the Renal Risk in Derby Study. It included 1741 older adult primary care patients with an average age of 73 years, who initially fell under CKD G3a or G3b based on two eGFR values obtained over more than 90 days. Utilizing the CKD-EPIcys, as well as combined equations to calculate eGFR, they observed significant changes in classification. Specifically, the use of cystatin C to confirm the diagnosis led to 7.7% of patients being classified as not having CKD, while 59% were identified with a more advanced stage of the disease. With the combined equation, 5.5% and 42.9% of patients were regrouped as not having CKD and presenting more advanced CKD, respectively [52]. 

In the Irish Longitudinal Study on Ageing, a cross-sectional analysis involving 5386 participants, researchers discovered a significant rise in cystatin C levels after the age of 65 years. This increase was so pronounced that the likelihood of a patient being reclassified as having more advanced CKD surged from 15% at age 50 years to 80% at age 80 years. These findings prompt the question of whether there are age-related factors that may complicate its role as a filtration marker. Moreover, studies have indicated that the addition of cystatin C to creatinine improves the accuracy of eGFR in patients aged over 70 years [19,53]. However, conflicting results were observed in patients aged over 80 years [19,54].

## 5. Conclusions and Future Perspectives

According to estimates from the Global Burden of Disease Study, all-age mortality attributed to CKD has risen to 41.5%, with diabetes emerging as a significant risk factor, accounting for half of all CKD-related deaths. 

CKD represents a serious economic burden and constitutes a key risk factor for cardiovascular diseases [1]. While serum creatinine remains the cornerstone biomarker due to its widespread availability and cost-effectiveness, cystatin C offers greater sensitivity due to less influence by factors such as muscle mass and diet. Future research should explore how cystatin C could be combined with other emerging biomarkers to enhance diagnostic accuracy, risk stratification, and therapeutic monitoring. For example, combining cystatin C with markers of inflammation, fibrosis, and cardiovascular health could provide a more holistic view of a patient’s condition and guideline more personalized treatment approaches. 

## Figures and Tables

**Table 1 ijms-25-08135-t001:** The percentage of undiagnosed CKD in patients with diabetes [13].

**Stage 1 (eRFG ≥ 90 mL/min/1.73 m^2^)**	98.9% undiagnosed
**Stage 2 (eRFG 60–89 mL/min/1.73 m^2^)**	95.1% undiagnosed
**Stage 3 (eRFG 30–59 mL/min/1.73 m^2^)**	82% undiagnosed
**Stage 4 (eRFG 15–29 mL/min/1.73 m^2^)**	47.1% undiagnosed
**Stage 5 (eRFG < 15 mL/min/1.73 m^2^)**	41.2% undiagnosed

**Table 2 ijms-25-08135-t002:** GFR categories in CKD [17].

GFR Category	GFR (mL/min per 1.73 m^2^)
G1	≥90 (Normal or high)
G2	60–89 (Mildly decreased)
G3a	45–59 (Mildly to moderately decreased)
G3b	30–44 (Moderately to severely decreased)
G4	15–29 (Severely decreased)
G5	<15 (Kidney failure)

**Table 3 ijms-25-08135-t003:** Albuminuria categories in CKD [17].

Category	Albumin Excretion Rate (mg/24 h)	Albumin-to-Creatinine Ratio (Approximately Equivalent)		Terms
		(mg/mmol)	(mg/g)	
A1	<30	<3	<30	Normal to mildly increased
A2	30–300	3–30	30–300	Moderately increased
A3	≥300	≥30	≥300	Severely increased

**Table 4 ijms-25-08135-t004:** Barriers in using cystatin C to estimate GFR [24].

Barriers in Using Cystatin C to Estimate GFR
Absence of education provided about cystatin C
Higher costs compared to serum creatinine
Lack of institutional guidelines and policies
Placement of cystatin C results within the electronic health record
Proficiency with serum creatinine versus unfamiliarity with cystatin C

**Table 5 ijms-25-08135-t005:** Classification of novel biomarkers [8].

**Glomerular biomarkers**	Ceruloplasmin, Fibronectin, Glycosaminoglycans, Immunoglobulin G, Laminin, L-PGDS, Serum cystatin C, Transferrin, Type IV collagen
**Tubular biomarkers**	KIM-1, L-FABP, RBP4, NGAL, Urinary cystatin C
**Biomarkers of inflammation**	Tumor necrotic factor-α, Tumor necrotic factor-α receptors, CTGF, IL-6, MCP-1, TGF- β
**Biomarkers of oxidative stress**	8oxodG, Pentosidine, Uric acid

Abbreviations: L-PGDS: lipocalin-type prostaglandin D synthase; KIM-1: kidney injury molecule-1; L-FABP: liver-type fatty acid-binding protein; RBP4: retinol-binding protein4; NGAL: neutrophil gelatinase-associated lipocalin; CTGF: connective tissue growth factor; IL-6: Interleukins-6; MCP-1: monocyte chemoattractant protein-1; TGF-β: transforming growth factor-Beta; 8oxodG: 8-Oxo-7,8-dihydro-2-deoxyguanosine.

## Data Availability

Data are contained within the article.

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
