# Peer review of "CYSTATIN C—A Monitoring Perspective of Chronic Kidney Disease in Patients with Diabetes"

_ijms, 2024, doi:10.3390/ijms25158135_

Round 1

Reviewer 1 Report

Comments and Suggestions for Authors

Suggestions for minor changes

-          Line 59: replace ‘elevated’. Suggestion: ‘abnormal’ (explanation: albuminuria becomes elevated, but GFR decreases in parallel with decreasing renal function)

-          Lines 159-161: reformulate the phrase ‘The most recent guide of the American Diabetes Association published in 2024 mentioned that the CKD-EPI Refit equation, which is now recommended for everyone is the eGFR formula’ Suggestion: ‘ …that the Refit 2021 CKD-EPI equation is  recommended for everyone for estimation of GFR’.

-          Line 192: change cystatin c in cystatin C

-          Lines 224-226: reformulate the phrase ‘The interest in using cystatin C as a marker for early CKD arises from the recognition that individuals with type 2 diabetes often transition through a prediabetic phase and may already exhibit renal dysfunction at the time of diagnosis’. Suggestion: ‘individuals with type 2 diabetes often transit through’ or ‘often go through a prediabetic…’

-          Lines 263-265: the phrase ‘Prolonged exposure to inflammation, oxidative stress, uremic toxins and calcium-phosphorus metabolism disorders increases the possibility of arterial stiffness and cardiovascular disease in patients affected by CKD.’ is a repetition. The same information is present (little differences) between lines 254-258: ‘Besides traditional risk factors like hypertension, smoking and aging, specific CKD-related factors such as uremic toxins (including advanced glycation end-products, phosphate, uric acid and endothelin-1) and mineral metabolism disorders can cause arterial stiffness and decreased  vascular compliance through chronic inflammation and oxidative stress.’

-          Line 265-266: the phrase ‘Briet et al. found that arterial stiffness increases as renal function declines in patients with mild-to-moderate CKD’ has no bibliography

-          Lines 268-269: insert bibliography [41] at the end pf the phrase ‘Luo et al. indicated that cystatin C has a positive correlation with brachial-ankle pulse wave velocity (r = 0.448, P < 0.001)’.

-          Lines 277-278: insert bibliography [23] at the end of the phrase ‘Yeli Wang et al. offer guidance on interpreting discordant eGFRcr and eGFRcys results in ambulatory adults.’

-          Lines 285-290: provide bibliography for both phrases in this paragraph

-          Lines 343-346: the phrase ‘In a phase 3 trial involving 4,745 patients with type 2 diabetes being treated with dapagliflozin …’ is not correct. Suggestion: ‘In a post hoc analysis that included data pooled from 9 phase 3, randomized clinical trials involving 4,745 patients with type 2 diabetes treated with dapagliflozin ..’ Also, the bibliography is not correct. Correct reference: [Mende C, Katz A. Cystatin C- and Creatinine-Based Estimates of Glomerular Filtration Rate in Dapagliflozin Phase 3 Clinical Trials. Diabetes Ther. 2016 Mar;7(1):139-51. doi: 10.1007/s13300-016-0158-y. Epub 2016 Feb 22. PMID: 26899432; PMCID: PMC4801818.]

-          Lines 346-350:  The correct bibliography for the whole paragraph ‘Furthermore, a meta-analysis of 11 population studies comprising 90,750 participants, which utilized eGFRcys-cr, demonstrated that eGFRcys enhanced the classification of CKD and improved the estimation of risk for all-cause mortality and end-stage kidney disease. Notably, for patients with CKD stage 3a (eGFR creatinine 45-59 349 mL/min/1.73m²), 42% were reclassified as having an eGFR > 60 mL/min/1.73m² when utilizing eGFRcys ; is: [Shlipak MG, Matsushita K, Ärnlöv J, Inker LA, Katz R, Polkinghorne KR, Rothenbacher D, Sarnak MJ, Astor BC, Coresh J, Levey AS, Gansevoort RT; CKD Prognosis Consortium. Cystatin C versus creatinine in determining risk based on kidney function. N Engl J Med. 2013 Sep 5;369(10):932-43. doi: 10.1056/NEJMoa1214234. PMID: 24004120; PMCID: PMC3993094.]

-          Line 360. Instead [19], correct bibliography is: [Shardlow A, McIntyre NJ, Fraser SDS, Roderick P, Raftery J, Fluck RJ, McIntyre CW, Taal MW. The clinical utility and cost impact of cystatin C measurement in the diagnosis and management of chronic kidney disease: A primary care cohort study. PLoS Med. 2017 Oct 10;14(10):e1002400. doi: 10.1371/journal.pmed.1002400. PMID: 29016597; PMCID: PMC5634538.]

-          Line 365: correct reference is only [46]

-          Lines 360-363: the entire information ‘A previous meta-analysis determined contrasting outcomes, indicating that a greater percentage of CKD G3a patients were reclassified to no CKD (ranging from 35% to 47%) than to a more advanced CKD stage (ranging from 21% to 27%). It's worth noting that the average age of the patients in this meta-analysis was notably younger, between 55 and 60 years old.’ is referring to the meta-analysis discussed in lines 346-348. Besides that it has no correct reference (correct: [Shlipak MG, Matsushita K, Ärnlöv J, Inker LA, Katz R, Polkinghorne KR, Rothenbacher D, Sarnak MJ, Astor BC, Coresh J, Levey AS, Gansevoort RT; CKD Prognosis Consortium. Cystatin C versus creatinine in determining risk based on kidney function. N Engl J Med. 2013 Sep 5;369(10):932-43. doi: 10.1056/NEJMoa1214234. PMID: 24004120; PMCID: PMC3993094.]), it would be better to reformulate something like: ‘In the meta-analysis of 11 studies performed by Shlipak MG et al, contrasting results were reported:…’ such as the reader understand it is the same reference/the same study

-          Lines 364-365 with the phrase ‘Additionally, variability in cystatin C levels with age was highlighted in the Irish Longitudinal Study on Ageing’ and lines 366-369 with  In a cross-sectional analysis involving 5,386 participants, researchers discovered a significant rise in cystatin C levels after the age of 65 years. This increase was so pronounced that the likelihood of a patient being reclassified as having more advanced CKD surged from 15% at age 50 years to 80% at age 80 years.’ shear the same source, the reference [46]. It would be better that the reader understand this. For example, continuing in the same paragraph, something like: Additionally, variability in cystatin C levels with age was highlighted in the Irish Longitudinal Study on Ageing; in this cross-sectional analysis involving 5,386 participants, researchers discovered a significant rise in cystatin C levels after the age of 65 years….

Author Response

Dear Reviewer,

We want to sincerely thank you for the detailed comments and suggestions you have given us regarding our manuscript. We have integrated the feedback received to correct errors and bring more coherence and consistency to the text. In addition, we reformulated some paragraphs according to your precise indications. These changes certainly improve the clarity of some sentences. Furthermore, we corrected and adapted all the references as you suggested.

These modifications are highlighted in green in the text.

We believe these revisions have significantly improved the accuracy and overall quality of our manuscript.

Thank you once again for your thorough review of our manuscript!

Kind regards,

Alexandra-Mihaela Visinescu

Reviewer 2 Report

Comments and Suggestions for Authors

In this review article the authors try to make a case for the use of Cystatin C for the assessment of renal function in diabetics.

The manuscript is well written addresses the important issue of reliable assessment of renal function in diabetics. 

Major Comments:

1. The authors propose that cystatin C and eGFRcys should be used more widely for the assessment of renal function in diabetics. Although they provide evidence that perhaps cystatin C is more sensitive, the reality is that despite its availability it is a test that still has not gained traction as a marker of renal function. The authors should expand on the barriers that have to a low use of this test at most centers at least in the US.

2. In addition, the authors should discuss on the availability and cost of the test and how it compares to creatinine. In many centers cystatin C is still a send out test which also likely contributes to its low use.

3. Although the review is focused on diabetes the authors should expand a bit more on the use and limitations of cystatin C in some settings (i.e cancer).

4. The authors should also touch on other biomarkers of diabetic nephropathy and how they correlate with cystatin C and creatinine

5. Given the number of studies they quote perhaps a table highlighting the most significant ones would be important

6. Not sure if the Figure depicting how the different equations have evolved is necessary

7. The authors should also discuss whether cystatin C should be used not only in diabetics but other causes of CKD

Author Response

Dear Reviewer,

Your comments were extremely valuable and helped us improve the quality of our research considerably.

We particularly appreciated the way you highlighted the areas that needed improvement. The recommendations allowed us to address certain issues that we had initially overlooked. Your input had a significant impact on the final quality of the paper and these revisions have significantly improved our manuscript.

All changes are highlighted in yellow in the text.

We hope that our revised submission meets your expectations and we thank you for your precious comments and suggestions!

Sincerely,

Alexandra-Mihaela Visinescu

Comment 1: The authors propose that cystatin C and eGFRcys should be used more widely for the assessment of renal function in diabetics. Although they provide evidence that perhaps cystatin C is more sensitive, the reality is that despite its availability it is a test that still has not gained traction as a marker of renal function. The authors should expand on the barriers that have to a low use of this test at most centers at least in the US.

Response 1: Thank you for highlighting the need to address the barriers to the wider adoption of cystatin C and eGFRcys for assessing renal function. We have expanded the discussion in our manuscript to include the important points regarding the low utilization of cystatin C testing.

Line 254: Table 4. Barriers in using cystatin C to estimate GFR [24]

Barriers in using cystatin C to estimate GFR

Absence of education provided about cystatin C

Higher costs compared to serum creatinine

Lack of institutional guidelines and policies

Placement of cystatin C results within the electronic health record

Proficiency with serum creatinine versus unfamiliarity with cystatin C

Comment 2: In addition, the authors should discuss on the availability and cost of the test and how it compares to creatinine. In many centers cystatin C is still a send out test which also likely contributes to its low use.

Response 2: Thank you for suggesting the inclusion of a discussion on the availability and cost of cystatin C testing compared to creatinine.

Lines 244-252: 2.6. Limitations in using cystatin C

The 2019 General Chemistry Survey by the College of American Pathologists indicated that only 7% of US clinical laboratories offered cystatin C tests and over 90% of these referred the testing to external commercial labs [24].

Cystatin C testing costs around GBP 2.50 (USD 3.00) per test, which is about 10 times higher than the GBP 0.25 (USD 0.30) cost of creatinine testing [38].

Cystatin C measurement costs more than creatinine testing, but it is less expensive than other common laboratory tests, such as brain natriuretic peptide, troponin, parathyroid hormone or vitamin D and significantly cheaper than mGFR testing [23]. 

Comment 3: 3. Although the review is focused on diabetes the authors should expand a bit more on the use and limitations of cystatin C in some settings (i.e cancer).

Response 3: Thank you for suggesting that we should expand our discussion on the use and limitations of cystatin C in other settings. We have included additional information in the manuscript regarding the use of cystatin C in other specific contexts as follows:

Lines 230-242: 2.5. Indications for the use of cystatin C

The current KDIGO guideline proposes the following indications for the use of cystatin C:

  • Diet: low-protein diet, keto diets, vegetarian, high-protein diets and creatine supplements;
  • llness other than CKD: malnutrition, heart failure, catabolic consuming diseases, cancer, cirrhosis, muscle wasting diseases;
  • Lifestyle: smoking;
  • Medication effects: broad spectrum antibiotics that minimize extrarenal elimination, decreases in tubular secretion, steroids (anabolic, hormone);
  • Variations in body habitus and muscle mass: eating disorders, extreme sports or exercise, bodybuilding, above-knee amputation, spinal cord injuries resulting in paraplegia, paraparesis, quadriplegia or quadriparesis and class III obesity [17].

Comment 4: The authors should also touch on other biomarkers of diabetic nephropathy and how they correlate with cystatin C and creatinine.

Response 4: Thank you for your valuable comment regarding the inclusion of other biomarkers of diabetic nephropathy and their correlation with cystatin C and creatinine. We have carefully considered your suggestion and included novel biomarkers.

Lines 259-268: 3. Novel biomarkers in diabetic kidney disease

Diabetic kidney disease is a heterogeneous condition with a complex pathophysiology. Therefore, it is unlikely that a single biomarker can predict its prognosis. A multi-marker approach may be necessary to predict disease progression [39].

Table 5. Classification of novel biomarkers [8]

Glomerular biomarkers

Ceruloplasmin, Fibronectin, Glycosaminoglycans,  Immunoglobulin G, Laminin, L-PGDS, Serum cystatin C, Transferrin, Type IV collagen

Tubular biomarkers

KIM-1, L-FABP, RBP4, NGAL, Urinary cystatin C

Biomarkers of inflammation

Tumor necrotic factor-α, Tumor necrotic factor-α receptors, CTGF, IL-6, MCP-1, TGF- β

Biomarkers of oxidative stress

8oxodG, Pentosidine, Uric acid

L-PGDS: lipocalin-type prostaglandin D synthase; KIM-1: kidney injury molecule-1; L-FABP: liver-type fatty acid-binding protein; RBP4: retinol-binding protein4; NGAL: neutrophil gelatinase-associated lipocalin; CTGF: connective tissue growth factor; IL-6: Interleukins-6; MCP-1: monocyte chemoattractant protein-1; TGF-β: transforming growth factor-Beta; 8oxodG: 8-Oxo-7,8-dihydro-2-deoxyguanosine

Comment 5: Given the number of studies they quote perhaps a table highlighting the most significant ones would be important.

Response 5: Thank you for your suggestion regarding the inclusion of a table to highlight the most significant studies mentioned in our manuscript. We understand the potential benefit of such a table in summarizing the most important studies, but each study has particularities that require detailed description to be properly understood. Reducing these details to a tabular format could lead to the loss of essential information and an incomplete understanding of the multiple advantages of the integration of cystatin C in the routine assessment of the patients with chronic kidney disease, especially when diabetes is associated.

Comment 6: Not sure if the Figure depicting how the different equations have evolved is necessary.

Response 6: Thank you for your valuable comment regarding the necessity of the figure depicting the evolution of different equations. We agree that this figure may not be essential to the the manuscript. Therefore, we have decided to remove it from the content of our article.

Comment 7: The authors should also discuss whether cystatin C should be used not only in diabetics but other causes of CKD.

Response 7: Thank you for your comment regarding the potential broader use of cystatin C in chronic kidney disease, beyond its application in patients with diabetes.

We agree that discussing the utility of cystatin C in various etiologies of chronic kidney disease is important for a comprehensive understanding. In response to your suggestion, we have included all the indications for the use of cystatin C (lines 230-242), comprising causes of chronic kidney disease according to the latest recommendations of Kidney Disease Involving Global Outcomes 2024. 

Round 2

Reviewer 2 Report

Comments and Suggestions for Authors

N/A